# Involvement of Oxidative Stress in Protective Cardiac Functions of Calprotectin

**DOI:** 10.3390/cells11071226

**Published:** 2022-04-05

**Authors:** Luc Rochette, Geoffrey Dogon, Eve Rigal, Marianne Zeller, Yves Cottin, Catherine Vergely

**Affiliations:** 1Equipe d’Accueil (EA 7460): Physiopathologie et Epidémiologie Cérébro-Cardiovasculaires (PEC2), Faculté des Sciences de Santé, Université de Bourgogne—Franche Comté, 7 Bd Jeanne d’Arc, 21000 Dijon, France; geoffrey.dogon@u-bourgogne.fr (G.D.); eve.rigal@u-bourgogne.fr (E.R.); marianne.zeller@u-bourgogne.fr (M.Z.); catherine.vergely@u-bourgogne.fr (C.V.); 2Service de Cardiologie, CHU-Dijon, 21000 Dijon, France; yves.cottin@chu-dijon.fr

**Keywords:** alarmins, calprotectin, inflammation, oxidative stress, myocardium

## Abstract

Calprotectin (CLP) belonging to the S-100 protein family is a heterodimeric complex (S100A8/S100A9) formed by two binding proteins. Upon cell activation, CLP stored in neutrophils is released extracellularly in response to inflammatory stimuli and acts as damage-associated molecular patterns (DAMPs). S100A8 and S100A9 possess both anti-inflammatory and anti-bacterial properties. The complex is a ligand of the toll-like receptor 4 (TLR4) and receptor for advanced glycation end (RAGE). At sites of infection and inflammation, CLP is a target for oxidation due to its co-localization with neutrophil-derived oxidants. In the heart, oxidative stress (OS) responses and S100 proteins are closely related and intimately linked through pathophysiological processes. Our review summarizes the roles of S100A8, S100A9 and CLP in the inflammation in relationship with vascular OS, and we examine the importance of CLP for the mechanisms driving in the protection of myocardium. Recent evidence interpreting CLP as a critical modulator during the inflammatory response has identified this alarmin as an interesting drug target.

## 1. Introduction

Inflammatory responses are important for fighting bacterial infections. Endogenous signals have the ability to indicate tissue damage and are therefore termed alarmins or damage-associated molecular patterns (DAMPs). The identification of alarmins as crucial mediators of the inflammatory processes in various disorders is demonstrated.

Among these alarmins, the S100 proteins are generally believed to be mediators of inflammation. In this family, calprotectin (CLP) is a heterodimeric complex formed by two binding proteins of the calcium ion. S100A8, S100A9 and CLP possess both anti-inflammatory and anti-bacterial properties. CLP is expressed as an anti-bacterial agent, mainly but not exclusively by neutrophils, when activated. Upon cell activation, CLP released extracellularly acts through DAMPs [1]. The signaling pathways of S100A8/A9 were demonstrated: S100A8/A9 is a ligand of the toll-like receptor 4 (TLR4) and of receptor for advanced glycation end (RAGE). CLP is a likely target for oxidation due to its co-localization with neutrophil-derived oxidants at sites of infection and inflammation. The extracellular effects of S100 proteins require binding to the receptor RAGE [2].

A large number of experimental data show that in the heart, neutrophils are the first immune cell population to respond to myocardial infarction (MI), promptly infiltrating the ischemic myocardium. In the acute inflammatory phase, neutrophils exacerbate the myocardial injury. In these conditions, S100A8 and S100A9 are abundantly stored in neutrophils as CLP and are rapidly released in response to inflammatory stimuli. Recent results indicate that oxidative stress (OS) responses and S100 proteins are closely related. They are closely linked in pathophysiological processes in the heart [3]. S100A9 blockade during the inflammatory phase post-myocardial infarction inhibits systemic and cardiac inflammation and improves cardiac function [4,5]. Our review summarizes the roles of S100 proteins (S100A8, S100A9 and CLP) in the inflammation in association with vascular OS. We also examine the importance of CLP for the driving mechanisms in the protection of myocardium during ischemic injury.

## 2. Structure and Expression of Calprotectin

In preamble, it is important to specify that S100A8 and S100A9 preferentially form a stable heterodimer in vivo. In addition, tetramer and heterotetramer formations were described; binding of divalent cations influence the conformational structure of these proteins. While S100A8 and S100A9 spontaneously form heterodimers in the absence of metal ions, in return the formation of tetramer forms is dependent of the presence of specific ions such as calcium Ca^2+^ and/or zinc.

CLP is the heterodimer of the proteins S100A8 and S100A9 (Figure 1). Each monomer is able to bind two calcium ions and can also bind other divalent metal ions [6]. CLP released extracellularly, acts as an alarmin. The best characterized alarmins in health and disease are S100 proteins and also high-mobility group protein B1 (HMGB1), high mobility group N (HMGN1), interleukins (IL) and heat shock proteins (HSPs). The S100s are low molecular weight proteins that represent the largest subgroup within the Ca^2+^-binding EF-hand, including more than 20 members. These proteins are characterized by an exclusive solubility in a solution of 100% saturated ammonium sulfate, which is termed “S100”. Synonyms have also been adopted, such as myeloid-related proteins-8 and -9 (MRP-8 and MRP-9), calgranulin A and B, or migration inhibitory factor-related protein of 8 and 14 kDa (MRP-8 and MRP-14). CLP subunits are low molecular weight proteins. The molecular masses of S100A8 and S100A9 are 10.8 and 13.2 KD, which include 93 and 114 amino acids, respectively. Each monomer of S100A8/S100A9 comprises 4 α-helices (HI, HII, HIII and HIV) and 2 loops (Loop I and Loop II) [7].

The different types of complexes of S100A8 and S100A9, e.g., homodimer, heterodimer and heterotetramer, are reported to be important for the biological functions of these proteins with specific intracellular and extracellular properties [8]. Interestingly, various activities of CLP appear to be connected to the ability of S100 proteins to bind various metal ions. Together with calcium-binding domains, CLP contains histidine-based, zinc-binding sequences. Similarly, as calcium, zinc also induces the formation of CLP tetramer [9]. In addition to the calcium-binding sites, there are two binding sites for transition metal ions at the A8/A9 heterodimer interface. The six transition metals, manganese (Mn), iron (Fe), cobalt (Co), nickel (Ni), copper (Cu) and zinc (Zn), are essential micronutrients to both hosts and bacterial pathogens.

Several studies have demonstrated that CLP binds various transition metal ions such as Zn(II), Cu(II) and Fe(II) with strong affinity, finally running to inhibition of bacterial growth [10]. The chelating properties are particularly important for the function of inflammatory-associated S100 proteins. S100A8 and S100A9 may inhibit microbial growth by chelating zinc. Zinc homeostasis has a major effect on immune cell function that can be cell-specific during infection. In response to pathogens, zinc depletion impairs phagocytosis of bacteria by neutrophils. Following phagocytosis, zinc is mobilized into the phagosome to addle the pathogen. In return, high levels of zinc in the phagosome could lower the capacity for the nicotinamide dinucleotide phosphate (NADPH) oxidase (NOX) to generate reactive oxygen species (ROS) [11].

It is possible that the ability of CLP to influence redox speciation plays a physiological role in metal homeostasis at the host–pathogen interface [12]. As we shall develop it further, iron is an essential ion playing key roles in many life processes. Heme and non-heme iron enzymes utilize this transition metal to carry out chemical transformations, including reactions involved in various metabolisms, DNA synthesis and defense against OS [13]. Extracellular functions of several S100s may be regulated by oxidative modification. Oxidative modifications of S100A8 and S100A9 can profoundly impact the functions of these proteins. The interplay between oxidative modifications and ion binding may be an important mechanism whereby intracellular functions of S100 proteins are regulated [14].

There are two major forms of CLP: secreted soluble form and in plasmatic membrane integrated form. S100A8 and S100A9 constitute approximately 40% of the cytosolic protein fraction in circulating human blood neutrophils and 1% in monocytes. Soluble form of CLP is produced by neutrophils and monocytes following stimulation. The soluble form of CLP can be found in various body fluids, i.e., plasma and urine [15]. It should be noted that in normal conditions, the concentration of CLP in human serum is less than 1 µg/mL, which increases in inflammatory diseases. The normal serum levels of CLP are estimated to range between 0.1 and 1.6 μg/mL, and can be elevated in numerous conditions such as infection, inflammation or cancer [16].

As we reported previously, CLP is a heterodimeric complex formed by two binding proteins of the calcium ion. Binding of calcium induces conformational changes, the so-called calcium saturated state. At higher concentrations, calcium induces aggregation of human CLP. Aggregated CLP interacts with the plasmatic membrane leading to cell death [17]. CLP binds Ca^2+^ which regulates the translocation of S100A8/A9 from cytosol to plasma membrane. In these conditions the translocation induces nicotinamide dinucleotide phosphate (NADPH) oxidases (NOX) activation, which subsequently causes enzymatic activation and the phagosomal process [18]. Next, we describe the relations between CLP activity, Ca^2+^ binding, NOX activation and redox balance.

## 3. Calprotectin Receptors: Toll-Like Receptors (TLRs) and Receptor for Advanced Glycation End (RAGE)

### 3.1. Interaction with “Alarm” Receptors: Toll-Like Receptor (TLRs)

TLRs recognize ligands derived from pathogenic organisms and are known as “pathogen-associated molecular patterns” (PAMPs) or host-derived DAMPs. TLRs are distributed throughout different tissues and respond with the release of endogenous substances such as S100 proteins. As we reported, S100 proteins were described as DAMP molecules with extracellular activity and interaction with “alarm” receptors such as TLRs and RAGE (Figure 1 and Figure 2). Importantly, TLRs may be constitutively expressed in some tissues such as on human umbilical vein endothelial cells (HUVEC). Upregulation of both receptors can be induced on the endothelial surface by various inflammatory stimuli [19]. S100A8/S100A9 complexes are released during activation of phagocytes and mediate their effects via TLR4, leading to the production of tumor necrosis factor alpha (TNF-α) and other cytokines [20]. Platelets express surface TLR4 and platelet-TLR4 is actively upregulated during most clinical forms of infectious and harmful inflammation. It has been reported that platelet-TLR4 was involved in the blood actions of lipopolysaccharide (LPS). Platelet TLR-4 activation by pathogen- or damage-associated molecular pattern molecules triggers pro-thrombotic, pro-inflammatory and pro-coagulant effector responses. Moreover, platelet TLR4 has a prominent role as a sensor of high LPS circulating levels during sepsis [21,22].

Endothelial cells are activated by microbial agonists through TLRs to express inflammatory mediators; this is of significance in inflammatory states such as septic shock. LPSs recruit interrelated redox process and kinase-dependent signaling pathways through endothelial TLR4 activation. Many studies demonstrated the function of LPS in the activation of endothelial TLR4 inducting a cascade of signaling pathways including: (1) NADPH oxidase/ROS/Endothelial nitric oxide (NO) synthase (eNOS) and (2) mitogen-activated protein kinase (MAPK) and nuclear factor kappa B (NF-κB); these pathways are involved in vascular functions [23,24]. A suggested mechanism includes the generation of ROS through NADPH oxidase and subsequent eNOS deactivation and decreased endothelial NO bioavailability. These various pathways lead to endothelial dysfunction [25].

### 3.2. Binding to the Receptor RAGE

#### 3.2.1. Extracellular Effects

As we reported previously, the extracellular effects of some S100 proteins require binding to the receptor RAGE (Figure 1 and Figure 2). RAGE has been reported to play key roles in chronic inflammatory conditions and host defense to infection. RAGE interacts with distinct families of ligands, mediating functions in extensive cell types such as cellular migration, proliferation and cell death. RAGE is a member of the immunoglobulin family of cell surface molecules recognizing ligands including advanced glycation end products (AGEs), amyloid-β-peptide and some S100 proteins. RAGE is expressed at low levels in most tissues and may be upregulated [26]. The surface receptor for RAGE and its soluble (sRAGE) and endogenous secretory (EN-RAGE) forms belong to the same superfamily. These receptors play major functions in inflammation and immunity, directly or through binding with AGEs and advanced oxidation protein products (AOPP). In these conditions, RAGE could be used as markers of disease severity [27,28].

#### 3.2.2. Intracellular Mechanisms

Recent results concerning the structure of RAGE and their endogenous interactions increase the comprehension of the mechanisms by which RAGE is implicated in pathological outcome. Intracellular mechanisms activated by RAGE interaction with ligands include the MAPKs, extracellular signal-regulated kinase (ERK)1/2, p38, c-Jun NH_2_-terminal kinase (JNK) and transcription factor NF-κB [29]. S100A8/A9 is co-localized with microtubules in activated monocytes; this interaction is dependent on Ca^2+^ concentration. S100A9 is phosphorylated in Ca^2+^-dependent manner supporting its translocation from cytosol to the cytoskeleton and membrane [7]. The intracellular activity of S100A8/A9 is regulated by a process of phosphorylation, but only the phosphorylated form of the complex is implicated in pro-inflammatory cytokine expression and secretion [30,31].

## 4. Calprotectin, Oxidative Stress and Redox Potential

### 4.1. Background on Oxidative Stress and Redox Potential

As we reported previously, various endogenous reactions are involved in biological redox reactions, including the reduction of molecular oxygen and the transport of oxygen. In this field, the role of free radical processes and redox-signalization in adaptation of the organism to changes in oxygen level is primordial. ROS and reactive nitrogen species (RNS: derived from NO) constitute the most important class of radical species generated in living systems. ROS are well recognized for their dual role as both deleterious and beneficial species. Under some circumstances, normal homeostatic regulatory mechanisms may be overwhelmed by the production of ROS, RNS and peroxynitrite (ONOO). Peroxynitrite is the product of the diffusion-controlled reaction between superoxide (O_2_^•−^) and ^•^NO. Because most radicals are short-lived species, they react quickly with neighboring molecules. Adaptation and response to environmental changes require dynamic and fast information distribution within the cell. In the body, metal ion excess or deficiency can potentially inhibit protein function [24,32]. Iron, an essential element for many important cellular functions in all living organisms, can catalyze the formation of free radicals. In the context of ROS and RNS, the concept of “oxidative stress” (OS) is frequently used in several biochemical, physiological and pathophysiological situations [33].

As we developed previously, transition metal ions are involved in many biological processes crucial for sustaining life. Iron limitation by the host was the first recognized form of nutritional immunity. During inflammation, increased production of hepcidin by the liver, neutrophils and macrophages inhibits iron secretion, thereby facilitating a reduction in free extracellular iron [34]. Iron exists mostly in the form of heme in the host. Heme is a necessary co-factor for NADPH oxidase, which is essential for producing ROS that elicit OS upon pathogens in the phagosome. The catabolism of heme by heme oxygenase-1 (HO-1) produces carbon monoxide (CO), which accesses the heme-containing respiratory complexes of bacteria [13]. To limit the unspecific reactive potential of transition metals, their availability in vertebrate hosts needs to be strongly regulated and especially limited during infection [35,36]. Divalent metals such as iron, zinc and also manganese (Mn) play major functions at the host–pathogen interface.

### 4.2. Effect of Calprotectin on Oxidative Stress and Redox Regulation

Considering the dual role of CLP in redox balance and oxidative stress (Figure 2), CLP is a target for oxidation due to its co-localization with neutrophil-derived oxidants at sites of infection and inflammation. Each CLP subunit contains a single cysteine residue, which in the heterodimer are in close proximity. The sulfur-containing amino acids cysteine and methionine are particularly susceptible to the actions of ROS and reactive chlorine species (RCS), which can damage proteins. Cysteine residues are inclined to oxidation, resulting in formation of sulfenic acid, which reacts with thiol groups to setup disulfides. When oxidants are produced by neutrophils, a disulfide-mediated cross-link between S100A8 and S100A9 subunits is able to arise at inflammatory sites [37,38].

Microbial growth studies demonstrate that iron depletion by CLP contributes to the growth inhibition of bacterial pathogens. CLP binds Zn and Mn via a His_3_Asp binding site; it thereby reduces the bacterial virulence and defense against OFR, resulting in major killing of bacteria by neutrophils [39]. One potential target of CLP-mediated metal deprivation is Mn-dependent superoxide defenses, which protect invading microbes from the oxidative burst induced by neutrophils [40]. Finally, many metal-sequestering proteins have pleotropic roles in the immune response by acting as a DAMP and/or opsonin during infection. Improved identification of the pleiotropic functions of metal-sequestering proteins is required to clarify the challenging interactions during initiation and development of the inflammation process [41]. The mechanisms of CLP must be comprehended in the context of the extracellular environment and information is missing concerning intracellular roles of CLP in metal homeostasis [39]. Elemental analysis of CLP-treated growth medium establishes that CLP reduces the concentrations of Mn, Fe and Zn [10].

### 4.3. Calprotectin, Oxidative Stress and Viral Infection

The contribution of OS and CLP to disease pathogenesis has also been explored in several viral infections. A relationship between OS and CLP levels was demonstrated in several respiratory viral infections. In this field, SARS-CoV-2 could trigger disruption of the balance between pro-oxidant and anti-oxidant mediators [42]. It has been reported that elevated CLP levels in peripheral blood cells could be used to discriminate severe from mild COVID-19 infection. Myeloperoxidase and CLP gene levels are increased in blood, lung tissue and inflammatory cells; SARS-CoV-2 induces the expression of OS genes via both immune and lung structural cells [43].

### 4.4. S100 Proteins, Oxidative Stress and Transient Receptor Potential

Studies have shown that S100 proteins play a role in chemotaxis and activation of immune cells in relationship with OS process and transient receptor potential (TRP). In endothelial cells, specific members of the TRP superfamily of cation channels act as important Ca^2+^ influx pathways [44]. They are involved in endothelium-dependent vasodilation and regulation of permeability and angiogenesis. ROS and RNS can modulate the activity of TRP channels mainly by modifying specific cysteine residues [45]. The importance of the redox regulation of TRP channel activity in endothelial cells in relation with S100 functions and the participation of these pathways for cardiovascular diseases has been reported [46]. TRPs channels can be modulated either by calcium itself and/or by calcium-binding proteins (CBPs). A CBP such as the S100 calcium-binding protein A1 (S100A1), is known for its modulatory activities toward receptors [47].

## 5. Role of Calprotectin on Chemotaxis and Trans-Endothelial Migration

As we reported, CLP abundantly stored in neutrophils was released in response to inflammatory stimuli and acted as damage-associated molecular outlines exerting intracellular and extracellular effects. This role of S100A8/A9 in promoting accumulation of leukocytes in the site of inflammation is supported by several studies [48]. Additionally, CLP involves a process of degradation of tight junctions among endothelial cells, facilitating the passage of cells in the site of inflammation [49]. In most tissues, the leukocyte recruitment cascade involves the following steps: tethering, rolling, adhesion, crawling and transmigration. Neutrophil recruitment is initiated by changes on the surface of endothelium that result from stimulation by inflammatory mediators such as cytokines. These mediators are released from tissue-resident leukocytes when they interact with pathogens. CLP promotes the accumulation of monocytes and monocyte-derived cells at the sites of inflammation, CLP participating on the extravasation of leukocytes [50].

Endothelial cells can also be activated directly by pattern-recognition receptor (PRR)-mediated detection of pathogens that increases expression of adhesion molecules. P-selectin and E-selectin, synthesized de novo, are upregulated [45]. These two selectins have partially overlapping functions and participate in neutrophil recruitment. Neutrophil infiltration is regulated through a complex sequence of molecular steps involving the selectins and the integrins, which mediate leukocyte rolling and adhesion to the endothelium. In the myocardium following infarction, the monocyte chemoattractant protein (MCP)-1 is also markedly upregulated in the infarcted myocardium, inducing recruitment of mononuclear cells in the injured areas. It has been reported that CLP induced MCP-1 production [51]. S100A8, S100A9 and S100A8/A9 have been shown to induce L-selectin shedding, neutrophil adhesion and macrophage-*1* antigen (Mac-1) activation [52]. Extracellular S100A8/A9 regulates leukocyte recruitment during the development of vascular disease by acting as a chemoattractant or by binding to cell receptors TLR4 and RAGE that activate leukocytes. Both receptors modulate vascular inflammation and participate in the pathobiology of atherosclerosis [53]. This S100A8/A9 further induces vascular cell adhesion molecules, resulting in reduced rolling velocity and faster adhesion for trans-endothelial migration [54].

## 6. Calprotectin—A Pleiotropic Molecule in Acute and Chronic Inflammation

The S100 family is commonly believed to be mediators of inflammation [55,56,57]. Regulations of pro- and anti-inflammatory functions of S100 family proteins are dependent on complex interactions including oxidation process and phosphorylation events [58].

### 6.1. Anti-Inflammatory Properties of S100A8 and S100A9 Proteins

S100A8 and S100A9 are implicated in anti-inflammatory roles in wound-healing and protection against oxidative tissue damage, the latter as a result of their capacity to scavenge oxidants. In the microenvironment of different diseases, these proteins are released by necrotic tissues when the tissue is damaged, the biological effects implicating specific receptors [59]. S100s may be modified by various post-translational modifications, including phosphorylation, methylation, acetylation and oxidation. It is increasingly evident that oxidative modifications of cysteine residues in S100s may be crucial in regulating their activities. Many redox-based signaling pathways are regulated by reversible alterations of the proteins. In this field, in the cellular microenvironment, S-nitrosylated proteins have been increasingly recognized as important determinants of many biochemical processes [60,61].

### 6.2. Modification of Proteins by S-Nitrosylation

Post-translational modification of proteins by S-nitrosylation is now considered as important as phosphorylation and can have positive or suppressive consequences. As S-nitrosylation is regulated by the local environment around cysteine residues, limited proteins are implicated in this modification. Concerning the S100A8 and S100A9 proteins, they are S-nitrosylated by NO donors including S-nitrosoglutathione (GSNO); it is important to remember that GSNO is the physiological regulator of NO transport and signaling [62]. S100A8-SNO complex has been detected in normal human neutrophils; it is increased significantly following activation. S100A8-SNO, is a stable NO adduct and NO transport transnitrosylated hemoglobin, suggesting its role in blood vessel homeostasis. Moreover, it repressed mast-cell-mediated activation and leukocyte adhesion to endothelial cells in microcirculation. In these conditions, S-nitrosylation of S100A8, S100A9 and CLP by NO generated by endothelial NOS may regulate blood flow during inflammation [60,63].

### 6.3. Anti-Inflammatory Properties of Calprotectin

Many studies using experimental models have revealed a beneficial role for CLP in controlling the inflammation process. It has been demonstrated that S100A8/A9 protected liver from LPS-induced inflammation in rats [64]. These proteins also reduced the inflammation process in experimental autoimmune myocarditis. Experimental autoimmune myocarditis was induced in rats by immunization with porcine cardiac myosin. The mRNA expression of IL-1beta, IL-6 and TNF-alpha in the myocardium of these immunized animals was significantly implicated in rats treated with S100A8/A9 complex. The findings of these experimental studies revealed high-affinity binding of S100A8/A9 with IL-1beta, IL-6 and TNF-alpha in the myocardium, implying the device of pro-inflammatory cytokines by S100A8/A9. Thus, treatment with S100A8/A9 is capable of neutralizing several pro-inflammatory cytokines [65]. As we reported previously, S100A8/A9 and CLP secreted by neutrophils, monocytes and macrophages during infection modulate the inflammatory process by binding to TLR4 or RAGE and activate the production of cytokines. CLP also facilitates anti-microbial properties of neutrophil extracellular traps (NETs) produced by neutrophils [66].

## 7. Calprotectin as Therapeutic Targets in Cardiac Disease

### 7.1. S100 Family Members and Cardioprotection

Following a myocardial infarction (MI), several S100 family members mediate pro-inflammatory cascades, which exert harmful effects in the infarcted heart [67,68]. Clinical studies reported that CLP is a protein whose abundance in circulation is significantly associated with the prognosis of acute coronary syndrome (ACS) and acute myocardial infarction (AMI). Experimentally, serum concentrations of S100 proteins such as S100B, S100A6 and S100P (a small, 10.4 kDa calcium-binding protein) are associated with the severity of MI in rat ischemia–reperfusion models and in clinic studies are increased in patients with ACS, particularly in those with ST-segment elevation MI (STEMI) [69]. Consequent to an MI, S100A8 and S100A9 are secreted mainly from activated myeloid cells (neutrophils and monocytes). This secreted CLP acts, as we previously reported, on TLR4 and RAGE receptors [70] of various cell types to activate signaling through NF-κB [71] and MAPK pathways. Cardiac fibroblasts also respond to CLP with increased synthesis of cytokines, proteases and extracellular matrix (ECM) proteins [72]. CLP binding to RAGE on cardiomyocytes leads to a decrease in Ca flux and a subsequent decrease in cardiomyocyte contractility [73].

Among the cardiac diseases, heart failure (HF) worsens patient life quality. This clinical situation is the end stage of compensative mechanisms due to the inability of the heart itself to maintain blood pressure for organ perfusion. After this initial decline in pumping capacity of the heart, a variety of compensatory mechanisms are activated. These mechanisms are associated with a vicious cycle resulting in maladaptive structure in failing hearts. Many proteins, including S100 proteins, have been implicated as some of the potentially biologically active molecules in HF [74]. S100A8/A9 complex plasma levels and other inflammatory biomarkers were significantly higher in chronic HF patients. A8/A9 complex, together with IL-6 and IL-8, was found to be increased in enrolled patients with chronic HF. These findings suggest a potential use of S100A8 and A9 as pro-inflammatory biomarkers adding sensitivity to the already existing markers [75,76]. Several lines of recent evidence support the idea that S100A8/A9 may exert cell-protective roles in cardiac hypertrophy. Myocardial hypertrophy is characterized by an increase in cardiomyocyte protein synthesis and cell volume involved in the transition from adaptive to maladaptive cardiac function. Based on experimental and clinical studies, a concept termed myocardial hypertrophic preconditioning has been proposed [77]. Short-term hypertrophic stimulation can render the heart resistant to subsequent hypertrophic stress, reducing the progression to HF. During this process, some genes may be specifically upregulated to either block hypertrophic signaling pathways or trigger atrophic signaling pathways. Specific genes induced during the regression of hypertrophy were identified [78]. Experimental studies showed that upregulation of S100A8/A9 following removal of transient hypertrophic stimulus contributes to the anti-hypertrophic and anti-HF effect of hypertrophic pre-conditioning [79]. S100A8 and S100A9 are likely to exert their intracellular regulatory activities by interacting with specific targets in a Ca^2+^-dependent manner and fatty acid metabolism. More recently, cardiac hypertrophy and remodeling have increasingly been documented as a consequence of activation of the immunologic process involving an inflammatory response implicated in the development of hypertrophy and its evolution to HF. The cardiovascular system is very sensitive to the action of the thyroid hormone (TH). One major cardiovascular manifestation of TH overload is cardiac hypertrophy. Recent results demonstrate that the S100A8/NF-κB signaling pathway is activated in cardiomyocytes following TH stimulation [80]. The mechanism involved in the cellular actions of S100A8 can be interpreted as a compensation process. Within the heart, the S100A8/S100A9 pathway contributes to the adaptive response to pressure overload by producing amphiregulin by cardiac macrophages, amphiregulin being an essential cardioprotective mediator [81,82].

### 7.2. Interactions between S100 Proteins—Arachidonic Acid and Cardioprotection

A wide range of cardioprotective factors in relationship with the properties of S100 proteins have been investigated to date, such as the cardioprotective role of fatty acids, specifically arachidonic acid (AA) (Figure 2). This fatty acid is present in the membranes of cells in an inactive state and can be released by phospholipases in response to several stimuli, such as ischemia. Moreover, AA represents a precursor of potent signaling molecules, i.e., prostaglandins through enzymatic and non-enzymatic oxidative pathways. AA can be nitrated by reactive nitrogen species leading to the formation of nitro-arachidonic acid (NO_2_-AA) [83]. In the endothelial area, the S100A8/A9 complex also binds to AA and polyunsaturated fatty acids in a Ca^2+^-dependent manner. Zn^2+^ and Cu^2+^ have an inhibiting effect on AA binding capacity of the S100A8/A9 protein complex [84].

AA is known to affect leukocyte adhesion by suppressing the expression of adhesion molecules on endothelial cells [85,86]. It has been also reported that AA directly binds to the TLR4 co-receptor, myeloid differentiation factor 2, and prevented saturated fatty acids from activating the TLR4 inflammatory signaling pathway. Similarly, AA reduced LPS-induced inflammation in macrophages and septic injury. Moreover, in myeloid cells, intracellular CLP may transfer AA to the NADPH oxidase complex, inducing oxidative burst, which further enlarges inflammation [87]. Following myocardial ischemia–reperfusion injury in mice, CLP is rapidly expressed and secreted by inflammatory cells (Figure 3) and fibroblasts which then induce pro-inflammatory signaling, leukocyte infiltration and cardiac dysfunction. AMI causes an abrupt rise in leukocyte numbers in blood and myocardium. Neutrophils, which are the first cells to infiltrate the infarcted myocardium, release CLP and other DAMPs that bind to pattern recognition receptors (PRRs) on the surface of various cell types, driving the expression of inflammatory cytokines. As we reported previously, these cytokines act to promote adhesive interactions between leukocytes and endothelial cells [5]. Finally, in the myocardium through the endothelial reactivity, CLP blockade administered during the inflammatory phase of the immune response is able to reduce cardiac inflammation, limiting myocardial damage and improving the cardiac function [88].

## 8. Conclusions and Perspectives

The knowledge accumulated thus far concerning the ongoing disease states indicated that many pathologies are associated with altered expression levels of S100 proteins. S100 family proteins are a more sensitive biomarker of inflammatory activity. Serum CLP concentration indicates severity of inflammation in systemic inflammatory response syndrome and sepsis [89]. CLP is also one of the proteins differentially expressed in Type 1 diabetes mellitus (T1DM) and has been used as a marker of metabolic diseases [90]. CLP accumulates in atherosclerotic lesions and a high serum level is a marker for carotid plaque density [91]. Antibodies targeting S100A8/A9, S100A4, S100A7 and S100P have demonstrated efficacy for several pathological conditions [92]. Taken together, in the cardiovascular and metabolic fields, CLP is not only a physiological actor in myocardial injury but is also a major prognostic biomarker [4]. An emerging concept in the area of myocardial protection is the progress of a multitargeted strategy to reduce the MI size. Pharmacological treatment of MI remains a challenge to overcome; this pathology is a leading cause of morbidity and mortality worldwide. A major advantage of the S100 proteins approach is that CLP therapy affects multiple targets influencing several myocardial intracellular signaling pathways and inflammatory cells.

## Figures and Tables

**Figure 1 cells-11-01226-f001:**
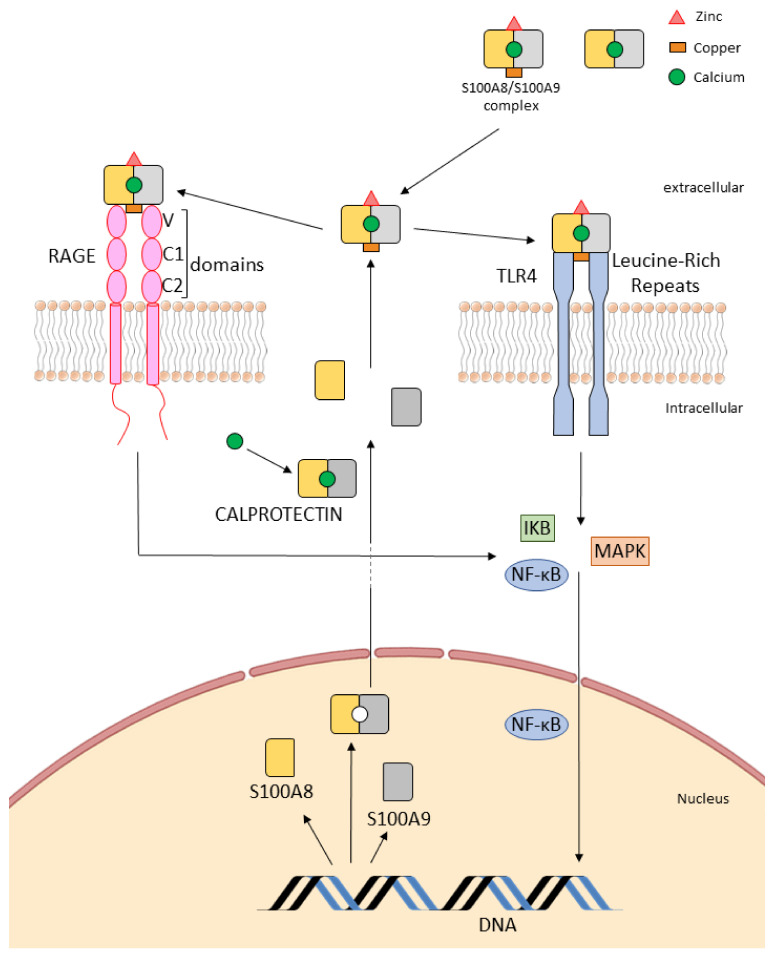
Schematic structure of S100A8/S100A9 complex, calprotectin (CLP), RAGE/TLR4 receptors and intracellular activity. The receptor for advanced glycation end products (RAGE) and toll-like receptor 4 (TLR4) are multiligand receptors. The ectodomain of RAGE is composed of three domains. The receptor is anchored in the plasma membrane by a single transmembrane helix and exhibits in the cytoplasm a short tail. Ligation of extracellular RAGE by S100A8/A9 complex promotes the homodimerization of RAGE. TLRs are type I transmembrane receptors characterized by an N-terminal extracellular domain, a single transmembrane domain and a C-terminal intracellular signaling domain. TLR4 exist as monomers in membrane before stimulation and dimerization induced by ligand recognition upon stimulation. CLP is a heterodimeric complex formed by two binding proteins of the calcium ion. CLP is the heterodimer of the proteins S100A8 and S100A9. Each monomer is able to bind calcium ions and can also bind other divalent metal ions such as zinc and copper. S100A8/A9 and CLP bind to RAGE and TLR4 receptors initiating signal transduction through NF-κB pathways and promoting increased S100A8/A9 and CLP transcription.

**Figure 2 cells-11-01226-f002:**
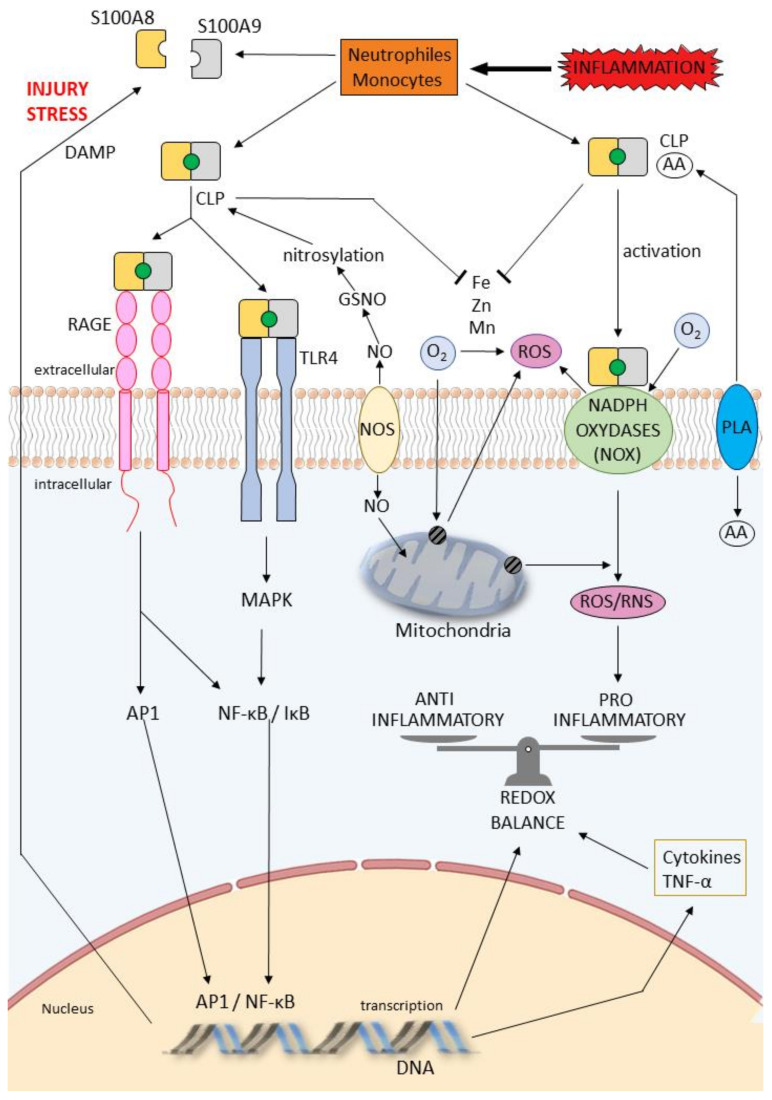
Signaling pathways of RAGE and TLR4 and schematic diagram of the contributory ROS and RNS induced by S100A8/A9 proteins, expressed and released in cells (neutrophils and monocytes) during injury stress. Both RAGE and TLR4 can trigger signaling events that activate the formation of endogenous complex, resulting in AP1/MAPK activation, NF-κB nuclear translocation and transcription of cytokines. Phospholipases A2 (PLA) catalyzes the hydrolysis of membrane phospholipids to liberate arachidonic acid (AA). In myeloid cells, intracellular CLP may transfer AA to the NADPH oxidase complex, inducing oxidative bursts that further enlarge inflammation. CLP binds various transition metal ions such as Zn(II), Cu(II) and Fe(II) with strong affinity, ultimately leading to reduction in ROS production in the mitochondria and through the oxidases. S100A8, S100A9 proteins and CLP are S-nitrosylated by NO donors including S-nitrosoglutathione (GSNO). NO is generated by endothelial NOS; AA: arachidonic acid; AP1: activator protein 1; CBP: calcium-binding proteins; CLP: calprotectin; DAMP: damage associated molecular pattern; MAPK: mitogen activated protein kinase; NADPH: nicotinamide dinucleotide phosphate; NF-κB: nuclear factor kappa B; NO: nitric oxide; NOS: nitric oxide synthase; NOX: nicotinamide dinucleotide phosphate (NADPH) oxidase; RAGE: receptor for advanced glycation end products; RNS: reactive nitrogen species; ROS: reactive oxygen species; TLR: toll-like receptor; TNF-α: tumor necrosis factor alpha; TRP: transient receptor potential; VCAM-1: vascular cell adhesion molecule 1.

**Figure 3 cells-11-01226-f003:**
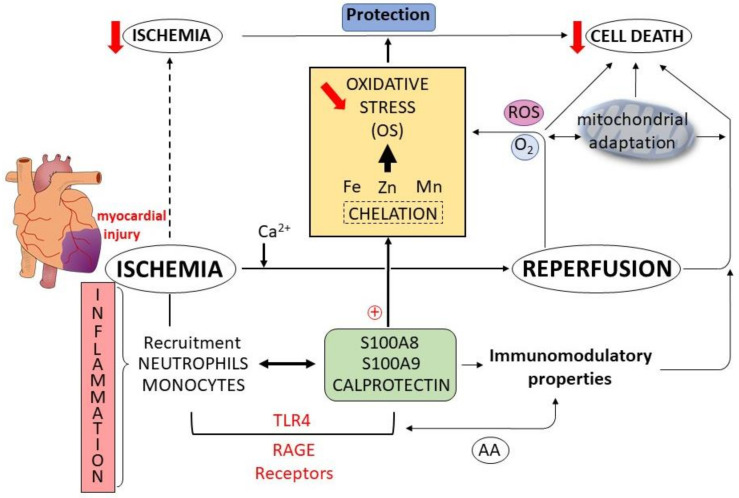
Schema showing the unifying mechanisms of cardioprotection induced by calprotectin (CLP) associated with oxidative stress (OS) modulation during ischemia–reperfusion sequence. This sequence induces inflammatory signaling, recruitment of neutrophils and monocytes associated with the potential development of cardiac dysfunction and cell death. Following myocardial ischemia injury, S100A8, S100A9 and CLP are expressed and secreted by inflammatory cells. S100 proteins and CLP bind to TLR-4 and RAGE receptors. Arachidonic acid (AA) possesses immunomodulatory properties, suppressing the expression of adhesion molecules on endothelial cells. CLP chelate transition metal ions (Fe, Zn, Mn) with strong affinity, leading to a reduction in OS, a mitochondrial adaptation and cardioprotection (reduction in ischemia). Red downward arrows: reduction of the process, red circled plus: positive activation of the process.

## Data Availability

Not applicable.

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
