# Peer review of "Involvement of Oxidative Stress in Protective Cardiac Functions of Calprotectin"

_cells, 2022, doi:10.3390/cells11071226_

Round 1

Reviewer 1 Report

The authors present an interesting review article examining the role of the calprotectin complex in injurious/inflammatory settings. Specifically, the authors highlight the complexing of neutrophil-derived calprotectin with the S100 proteins at the site of cellular injury, underlining the importance of oxidative processes in influencing the pro- and anti-inflammatory balance in injurious settings. Taken together, the influence of this paradigm appears to be incredibly important in the context of vascular-associated disease states, and presents opportunities to develop strategic interventions for ischemic/reperfusion-associated injuries.

In reviewing the manuscript, I had a number of concerns. The following should be addressed when preparing a suitable revision.

  1. There are a number of typos and formatting errors throughout the manuscript in both the writing and figures. While the information is easily read, the abundance of these errors is at times distracting, and this needs to be improved upon in any resubmission.
  2. The frequency of referencing could be improved. There are at times large blocks of text which depend on a single reference.
  3. Moreover, the type of references used needs to be improved upon. In reviewing the information, several of the articles references are indeed reviews themselves. In this way the information presented is at times quite shallow, and sections tend to gloss over details without ever going into any kind of depth on the role these pathways play in this context of the articles focus. This very much needs to be improved upon.
  4. The diagrams in general are very good but in Figure 3 some labels could be improved in terms of size and also the resolution – it is difficult to determine what exactly is being conveyed in certain steps.   

Reviewer 2 Report

I considered the manuscript entitled “Involvement of Oxidative Stress in Protective Cardiac Functions of Calprotectin” by Luc Rochette, et al, that is intended to be published in Cells.

I carefully read the manuscript and found a lot of information about calprotectin, basic and translational. Its properties, scenarios where it acts, mechanisms, interactions…. It is ok and comprehensive. Some paragraphs are repetitive or redundant and should be lighten up. However, information concerning the myocardium is scarce, as there is no more in the literature. So, much ado about nothing.

There are minor concerns to be repaired:

Line 159: chronic inflammatory states such as septic shock. Septic shock is an acute illness

Line 398: expend?

Finally, Conclusions and perspectives section is little informative. There are several sentences and literature references which are unconnected. It must be rewritten and shortened, focusing on the idea of the title, and introducing a clear message or reflections.

Round 2

Reviewer 1 Report

The authors have made an effort to address my comments and as such the new draft is much improved. However I do still have concerns in that the grammar of the piece still has instances that could be improved upon, and also I would have liked to have seen the inclusion of more data-orientated research articles to support certain sections. While improved, some of my original critique still stands. 
